# FUNCTIONAL RISK MINIMIZATION

## ABSTRACT

In this work, we break the classic assumption of data coming from a single function $f_{\theta^*}(x)$ followed by some noise in output space $\mathcal{P}(y|f_{\theta^*}(x))$. Instead, we model each data point $(x_i, y_i)$ as coming from its own function $f_{\theta_i}$. We show that this model subsumes Empirical Risk Minimization for many common loss functions and captures more realistic noise processes. We derive Functional Risk Minimization (FRM), a general framework for scalable training objectives that results in better performance in supervised, unsupervised, and reinforcement learning experiments. We also show that FRM can be seen as finding the simplest model that memorizes the training data, providing an avenue towards understanding generalization in the over-parameterized regime.

## 1 INTRODUCTION

### 1.1 MOTIVATION

In most machine learning settings, we only have limited control over how data is collected and even less so over the process generating it. For this reason, data is often correlated in complex ways, like data coming from similar times or locations. When these correlations are known, one can handle them appropriately as is done in frameworks such as multi-task or meta learning. However, in the absence of obvious reasons to specialize models to subsets of the data, practitioners often take an opposing perspective where differences in labels belonging to similar inputs are regarded as *noise*, often modeled in the output space. This idea serves as the basis for the training objectives we prefer, e.g., mean-squared error objective for gaussian noise or cross-entropy objective for multinomial distributions. By not accounting for highly structured noise, we expect that a singular model will appropriately average out noise differences during training.

For instance, consider training a language model on Wikipedia, then fine-tuning it to work on a dataset of books. In doing so, we use two different functions $f_{\theta_{\text{books}}}$ and $f_{\theta_{\text{wiki}}}$ with $f_{\theta_{\text{books}}} \approx f_{\theta_{\text{wiki}}}$. In contrast, when we train a model on general internet data, using Wikipedia and the dataset of books, we typically use a single function $f_{\theta_{\text{internet}}}$, and we explain each training example with multinomial noise in output space, i.e., $y_i \sim \mathcal{P}(\cdot|f_{\text{internet}}(x_i))$. However, whether we arrange the data into different datasets or a single one, the datapoints remain the same. Therefore, it is contradictory to handle the same variability using two different models: functional diversity vs. output noise.

To remedy this contradiction, this paper proposes to model noise in function space instead of output space. We propose **Functional Generative Models (FGMs)**, where each point $(x_i, y_i)$ comes from its own (unseen) function, $f_{\theta_i}$, which fits it: $y_i = f_{\theta_i}(x_i)$. FGMs don't assume the existence of a privileged function $f_{\theta^*}$, but consider a distribution over functions $\mathcal{P}(\theta)$, see fig. 1.

Most supervised machine learning is based on variants of empirical risk minimization (ERM), which searches for a single function that best fits the training data. There, the training objective acts in output space, comparing the true answer with the prediction. In contrast, assuming that data comes from an FGM, we derive the **Functional Risk Minimization (FRM)** framework, where training objectives act in function space. Although the full version requires a high-dimensional integral, we derive a reasonable approximation that scales to training neural networks.

Recently, neural networks have been observed to generalize despite memorizing the data, contradicting the classic understanding of ERM (Zhang et al., 2017). Interestingly, we find a connection between FRM and a recent theory explaining this *benign overfitting* of over-parameterized neural networks under ERM.

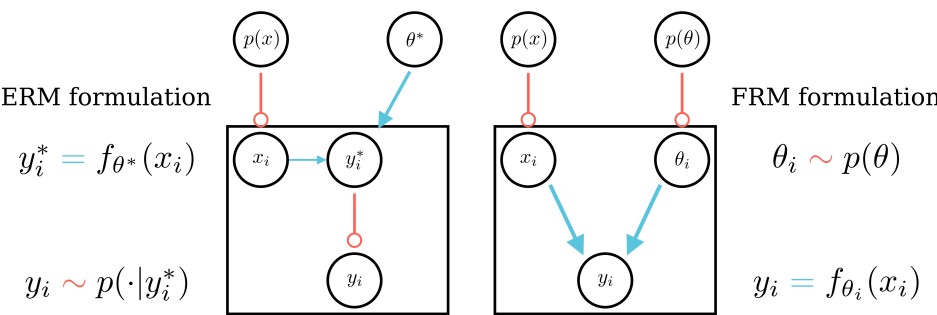

Figure 1: For many common losses, ERM and FRM can be related to maximum likelihood under simple generative models. Red lines ending in a circle are stochastic, blue arrows are deterministic.

The main contributions of this work are the following:

1. We introduce Functional Generative Models, a simple class of generative models that assigns a function to each datapoint.

2. We derive the Functional Risk Minimization framework, compute a tractable and scalable approximation and link it to the generalization of over-parameterized neural networks.

3. We provide empirical results showcasing the advantages of FRM in supervised learning, unsupervised learning, and reinforcement learning.

## 2 BACKGROUND AND RELATED WORK

### 2.1 INFERENCE AND RISK MINIMIZATION

In parametric machine learning, the user specifies a dataset $\mathcal{D} = ((x_i, y_i))_{i=1}^n$, a parameterized function class $f_\theta$, and a loss function $\mathcal{L}(y, f_\theta(x))$. Our goal is to design a learning framework that provides the $\hat{\theta}$ that minimizes the expected risk over unseen data: $\min_{\hat{\theta}} \mathbb{E}\left[\mathcal{L}\left(y, f_{\hat{\theta}}(x)\right)\right]$. However, since we do not have access to unseen data, we cannot compute this expectation.

**Empirical risk minimization (ERM)** In machine learning, we often rely on variants of ERM where a loss function $\mathcal{L}$ evaluated on the given dataset is optimized, i.e., $\min_\theta \sum_{i=1}^n \mathcal{L}(y_i, f_\theta(x_i))$. However, what we want to have is low *expected* risk (test loss), not empirical risk (training loss). In general, the best choice for a training objective depends on the loss function $\mathcal{L}$, but also on the (known) functional class $f_\theta$ and the (unknown) data distribution $\mathcal{P}(x, y)$. Often, ERM can be seen as doing maximum likelihood by assuming a very particular noise model for the data that makes $\mathcal{P}(y|x)$ a function of $\mathcal{P}(y|f_{\theta^*}(x))$ for some unknown, but fixed, $\theta^*$. However, in general, the user-defined loss function $\mathcal{L}$, and thus the optimal $\theta^*$, need not have any relation to the data distribution.

**Bayesian learning** The Bayesian setting explicitly disentangles inference of $\mathcal{P}(y|x)$ from risk minimization of $\mathcal{L}$. However, it usually assumes the existence of a true $\theta^*$, and further assumes it comes from some known prior $q$: $\theta^* \sim q(\cdot)$. Then, similar to maximum likelihood, the Bayesian setting often assumes a noise model $\mathcal{P}(y|f_{\theta^*}(x))$ on the output. Thus inference about the posterior, $\mathcal{P}(\theta|\mathcal{D}) \propto q(\theta) \cdot \mathcal{P}(\mathcal{D}|\theta)$, becomes independent of the loss. Only in the final prediction step, the loss function is used, together with the posterior, to find the output with the lowest expected risk.

**Relations to FRM** Similar to Bayesian learning, Functional Risk Minimization benefits from a clean distinction between inference and risk minimization. However, FRM assumes fundamental aleatory noise in function space $\mathcal{P}(\theta)$, not to be confused with epistemic uncertainty in the Bayesian setting. Similar to ERM, FRM aims at only using a single parameter $\theta^*$ at test-time, which avoids the challenging integration required in the Bayesian setting and its corresponding inefficiencies.

### 2.2 RELATED WORK

FGMs essentially treat each individual point as its own task or distribution. In this way, FGMs are related to multi-task learning (Thrun & Pratt, 1998) and meta-learning (Hospedales et al., 2020). Within them, connections between learning to learn and Hierarchical Bayes are the most relevant (Tenenbaum, 1999; Griffiths et al., 2008; Grant et al., 2018). Implementation-wise, FRM is closer to works looking at distances in parameter space (Nichol et al., 2018) or using implicit

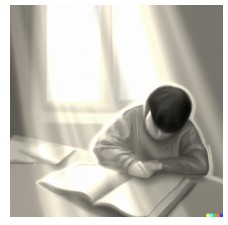

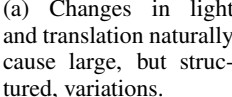

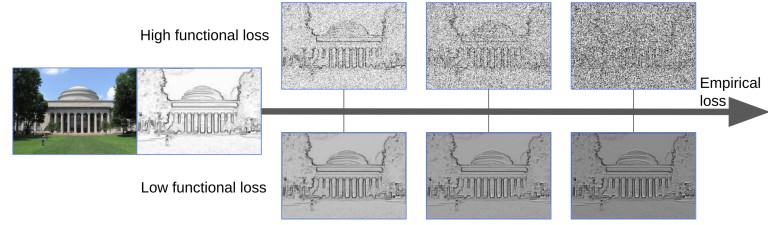

(a) Changes in light and translation naturally cause large, but structured, variations.

(b) Images with high and low functional loss for a series of fixed empirical losses, when predicting the edges of the image on the left. The model is a simple two-layer fully-convolutional network. One can see that images with low functional loss retain most of the structure despite having high errors in output (pixel) space.

Figure 2: Functional losses provide a way to capture structured noise, typical in natural settings.

gradients (Lorraine et al., 2020; Rajeswaran et al., 2019). However, these are still fundamentally ERM-based as noise is modeled in output space within each task.

Other works have noted the importance of function space for applications such as minimizing catastrophic forgetting in continual learning (Kirkpatrick et al., 2017), optimization (Martens & Grosse, 2015), or exploration in reinforcement learning (Fortunato et al., 2017). Information geometry (Amari, 2016), formalizes the geometrical structure of distributions using tools from differential geometry. In contrast, we leverage stochasticity in function space for modeling and learning.

Multiple alternatives to ERM have been proposed, particularly in the multi-task setting, such as adaptive (Zhang et al., 2020) and invariant risk minimization (Arjovsky et al., 2019). It is also relevant the line aiming at flat minima (Hochreiter & Schmidhuber, 1997)/minimizing sharpness (Foret et al., 2020) in order to improve generalization on standard supervised learning. In contrast to these works, our perturbations are per-point, and they come from the data distribution giving rise to the noise, instead of a regularization made on top of ERM with classic loss functions. Other works proposed per-point adaptations to *tailor* a model to each specific input either to encode an inductive bias (Alet et al., 2020; 2021) or adapt to a new distribution (Sun et al., 2019; Wang et al., 2020). However, these adaptations fine-tune an imperfect model trained with ERM to get it closer to an ideal model. In contrast, in this work, per-point models are not a mechanism, but a fundamental part of reality, which then defines losses in function space rather than output space.

## 3 FUNCTIONAL GENERATIVE MODELS: SAMPLING PER-POINT FUNCTIONS

### 3.1 DESCRIPTION

In machine learning, we want to reach conclusions about a distribution $\mathcal{P}(x,y)$ from a finite dataset $((x_i, y_i))_{i=1}^n$. However, there is no generalization without assumptions. From convolutions to graph neural networks and transformers, most research has focused on finding the right inductive biases for the mappings $x \mapsto y$. However, much less research has challenged the assumptions about the uncertainty of those mappings: $\mathcal{P}(y|x)$. For instance, whenever we minimize mean-squared error on an image-prediction problem we are doing maximum likelihood assuming gaussian noise in pixel space. However, the actual noise is usually much more structured, as we show in figure 2.

In this work, we start from a single principle, which we call Functional generative models (FGMs): we model each data-point $(x_i, y_i)$ as coming from its own function $f_{\theta_i}$ such that $y_i = f_{\theta_i}(x_i)$ and $\theta_i \sim \mathcal{P}(\theta)$. Notably, $\mathcal{P}(\theta)$ is unknown in the same way that we do not know $\mathcal{P}(x,y)$. FGMs can be seen as a special type of hierarchical Bayes (Heskes, 1998; Griffiths et al., 2008), where each group has a single point, the lower-level is deterministic and each $\theta_i$ is an unobserved latent variable.

**Example: predicting house prices with linear regression** Let's consider predicting the price of a house given its surface area using a linear regressor: $y = \lambda x + \beta$ and the mean-squared error loss function. ERM would simply find the $\lambda, \beta$ leading to the lowest squared error on the training data. This is equivalent to doing maximum likelihood on a gaussian noise model $y_i \sim N(\lambda x_i + \beta, \sigma^2)$ with constant $\sigma$. However, this may be suboptimal. For instance, we intuitively know that prices of bigger houses tend to be higher, but also have larger variances: we expect the price of a large house to vary by 500k, but we would not expect the same 500k variation for a small house.

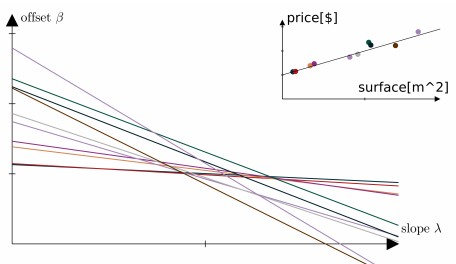 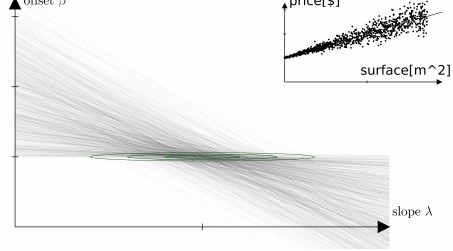

(a) Functional subspaces (lines in this example) that fit each point in the dataset (top plot). Each line in is colored according to its datapoint.

(b) The best parameter distribution (in green) being quite certain in the offset $\beta$, and uncertain in the slope $\lambda$.

Figure 3: Functional generative models for a linear function class in house price prediction. Since we only have two parameters, we can plot the function space in 2D on the bottom-left of each sub-figure, with the actual data is plotted on the top-right.

When using the FRM framework, we assume that, for each house $(x_i, y_i)$ there are different $\lambda_i, \beta_i$, satisfying $y_i = \lambda_i x_i + \beta_i$. For instance, we may believe that agent commissions vary and are well-modeled by $\beta_i$, and that the price-per-meter-squared (captured by $\lambda_i$) changes depending on the neighborhood. This is the modeling made by FGMs, which is more flexible than the output-level noise model corresponding to mean-squared error. We show this effect in figure 3.

## 3.2 PROPERTIES OF FGMs

**FGMs model the arbitrariness of dataset definitions**  A dataset implicitly defines which points belong to the data distribution $\mathcal{P}(x, y)$ and which points do not. For instance, a dataset of houses sold in Boston in the last 5 years, doesn't contain houses sold in other cities, or Boston houses sold in 2005. Each of these categories would follow a slightly different distribution and, using Hierarchical Bayes, we could model them as similar parameter assignments to a single function class.

More subtly, even the dataset of a single city encompasses multiple distributions, such as houses from different neighborhoods, years, or colors. These hidden intra-distributions are a source of noise when not described in the input. In the absence of any information, the least restrictive assumption is that each point comes from its own distribution, giving rise to what we refer as noise. It is therefore natural to use Hierarchical Bayes to model the differences in $\mathcal{P}(y_i|x_i)$ from a single $\theta_i \sim \mathcal{P}(\theta)$.

**FGMs entrust what the user already trusts**  A user needs to provide a learning framework with three ingredients: a dataset $((x_i, y_i))_{i=1}^n$, a function class $f_\theta$, and a loss function $\mathcal{L}$. Compared to the Bayesian setting, FGMs don't assume an independent noise model, which may have little connection with the user specifications. Instead, they leverage the user's trust in the function class $f_\theta$ to be a good model of the mapping $x \mapsto y$. They simply go one step further and also entrust the uncertainty in that mapping to the same function class, which now also models individual mappings $x_i \mapsto y_i$.

**FGMs encode structure through their function class**  FGMs draw their representational power from the function class $f_\theta$. Therefore, if the function class has a particular constraint, the FGM will have a corresponding constraint in probability space. For example, for the function class of linear functions, the expectation of $\mathcal{P}(y|x)$ is also linear. Similarly, as shown in figure 2, using convolutional neural networks we can create meaningful, structured noise priors in image space. From graph neural networks and neural differential equations to probabilistic programs, FGMs leverage structured function maps to construct structured probability distributions.

**FGMs can be arbitrarily expressive**  FGMs assume that $\mathcal{P}(y|x) = \mathbb{P}_{\theta \sim \mathcal{P}(\theta)}[f_\theta(x) = y]$. As just described, this need not be arbitrarily expressive. However, for some arbitrarily expressive function classes, such as multi-layer perceptrons, their corresponding FGM can be shown to be arbitrarily expressive, in probability space. We formalize this in the following definition.

**Definition 1.** Given a function class $\mathcal{F}$ with parameterisation $\Theta$, we define a Functional generative model $(\mathcal{P}(x), \mathcal{P}(\theta)) \in FGM[\mathcal{F}_\Theta, \mathcal{X}]$ as a probability density function $\mathcal{P}(x, y) \in L^2[\mathcal{X} \times \mathcal{Y}]$ with $x \sim \mathcal{P}(x) \in L^2[\mathcal{X}]$, and $y \sim \delta(f_\theta(x)), \theta \sim \mathcal{P}(\theta) \in L^2[\Theta]$.

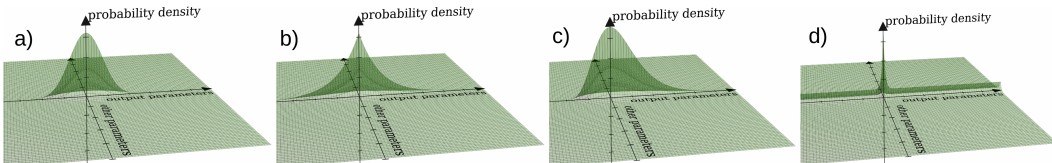

Figure 4: ERM with common losses is equivalent to maximum likelihood under an FGM that is only stochastic in the output parameters. The particular distribution depends on the loss: a) MSE with a Gaussian b) L1 with a Laplace c) cross-entropy with a Gumbel d) accuracy with a delta plus flat distribution. In practice, the axis for "other parameters" will often refer to thousands of parameters. Note that $\mathcal{P}(\theta \in \Theta)$ and $\mathcal{P}(x \in \mathcal{X})$ are independent and $y$ is deterministic given $x, \theta$; see figure 1.

**Theorem 1** (**Universal Distribution Theorem**). *Let $q(x, y) \in L^2[\mathcal{X} \times \mathcal{Y}], \mathcal{X} = [0, 1]^n \subset \mathbb{R}^n, \mathcal{Y} = [0, 1]^m \subset \mathbb{R}^m$ be a given probability density distribution function. Let $\mathcal{F}_{\Theta}^k$ be the class of 3-layer neural networks with sigmoidal activation function and $k$ neurons in the hidden layer. For any $\epsilon > 0$, $\exists K$ and a functional generative model $(\mathcal{P}(x), \mathcal{P}(\theta)) \in FGM\left[\mathcal{F}_{\Theta}^K, \mathcal{X}\right]$ s.t. $D_{TV}\left((\mathcal{P}(x), \mathcal{P}(\theta)), q\right) < \epsilon$, with $D_{TV}$ being the total variation distance. [Proof in appendix C.]*

**FGMs is a superset of some instances of ERM** In appendix B we prove that ERM for four common objectives (MSE, L1 loss, accuracy and cross-entropy) can be seen as a subcase of maximum likelihood on an FGM where all the stochasticity is restricted to the 'output' parameters. Figure 4 provides a visual intuition on how empirical losses correspond to functional losses in output space.

## 4 FUNCTIONAL RISK MINIMIZATION: LEARNING IN FUNCTION SPACE

Now, we look at the supervised learning problem under the FGM assumption.

### 4.1 MATCHING PROBABILITY DISTRIBUTIONS IN FUNCTION SPACE

We start with our goal to minimize the expected risk, impose the FRM generative model assumption and do basic math manipulations. In the derivation, whenever we use $\mathcal{P}(\theta)$ we refer to an unknown probability distribution entirely characterized by the data distribution $\mathcal{P}(x, y)$ and function class $f$.

$$\arg\min_{\theta^*} \mathbb{E}_{x,y}\left[\mathcal{L}(y, f_{\theta^*}(x))\right] = \quad (1)$$

$$\arg\min_{\theta^*} \int_x \int_\theta \mathcal{L}\left(f_\theta(x), f_{\theta^*}(x)\right) \mathcal{P}(\theta)\mathcal{P}(x) d\theta dx = \quad (2)$$

$$\arg\min_{\theta^*} - \int_\theta \mathcal{P}(\theta) \log\left(e^{-\int_x \mathcal{L}(f_\theta(x), f_{\theta^*}(x))\mathcal{P}(x)dx}\right) d\theta = \quad (3)$$

$$\arg\min_{\theta^*} - \int_\theta \mathcal{P}(\theta) \log\left(e^{-\mathbb{E}_x \mathcal{L}(f_\theta(x), f_{\theta^*}(x))} \cdot \frac{Z(\theta^*)}{Z(\theta^*)}\right) d\theta = \quad (4)$$

$$\arg\min_{\theta^*} H\left(\mathcal{P}(\theta), \frac{e^{-\mathbb{E}_x \mathcal{L}(f_\theta(x), f_{\theta^*}(x))}}{Z(\theta^*)}\right) - \log\left(Z(\theta^*)\right) = \quad (5)$$

$$\arg\min_{\theta^*} H\left(\mathcal{P}(\theta), \mathcal{Q}_{\theta^*}(\theta)\right) - \log\left(Z(\theta^*)\right). \quad (6)$$

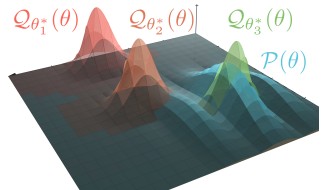

Figure 5: Finding the projection of the unknown distribution $\mathcal{P}(\theta)$ to the family $\mathcal{Q}_{\theta^*}(\theta)$ of probability distributions in function space. Here $\theta_3^*$ (green) is best.

with $H(\mathcal{P}, \mathcal{Q})$ being the H cross-entropy operator and $\mathcal{Q}_{\theta^*}(\theta) = e^{-\mathbb{E}_x \mathcal{L}(f_\theta(x), f_{\theta^*}(x))}/Z(\theta^*)$, $Z(\theta^*) = \int_\theta e^{-\mathbb{E}_x \mathcal{L}(f_\theta(x), f_{\theta^*}(x))} d\theta$ being a class of probability distributions and their normalizers.

To gain some intuition, we first observe that the second term $-\log Z(\theta^*) = \log 1/Z(\theta^*) = \log 1/\left(\int_\theta e^{-\mathbb{E}_x \mathcal{L}(f_\theta(x), f_{\theta^*}(x))} d\theta\right)$ is a label-independent regularizer that penalizes $\theta^*$ leading to small $\int_\theta e^{-\mathbb{E}_x \mathcal{L}(f_\theta(x), f_{\theta^*}(x))} d\theta$; i.e. a sharp distribution. Now, we can see that the first term is encouraging the matching of two probability distributions in function space:

1. $\mathcal{P}(\theta)$: the unknown data-dependent distribution, which does *not* depend on the loss function $\mathcal{L}$. This target distribution is defined entirely by the model class $f$ and the unknown data distribution $\mathcal{P}(x, y)$, which we will have to estimate from the training data.

2. $\mathcal{Q}_{\theta^*}(\theta)$: a class of probability distributions which depends on the loss function $\mathcal{L}$ and the $\theta^*$ used to make predictions, but not on the labels. This approximating distribution makes a parameter $\theta$ more likely the closer the function $f_\theta$ is to $f_{\theta^*}$ according to the problem-specified loss $\mathcal{L}$. Intuitively, it is a gaussian-like distribution centered at $\theta^*$, with a metric that captures the differences in task space. This will be formalized in section 4.2.

This equation also shows that we need *not* know the exact shape and distribution of $\mathcal{P}(\theta)$, which could be very complex without further assumptions. We only need to know its 'projection' to a particular class of probability distributions defined by the task at hand. This also happens in ERM-based learning: we need not know $\mathcal{P}(y|x)$ in order to estimate a $x \mapsto y$ map.

We would like to optimize equation 6, but we do not have access to samples for $\mathcal{P}(\theta)$, we only have $(x, y)$ pairs. However, we can compute the cross-entropy on $\mathcal{P}(y|x)$ following the FRM generative model. Thus, for a given dataset $\mathcal{D}_{train} = ((x_i, y_i))_{i=1}^n$ the FRM objective is:

$$\arg\max_{\theta^*} \sum_{(x_i, y_i)} \log \int_{\theta_i : f_{\theta_i}(x_i) = y_i} e^{-\mathbb{E}_x\left[\mathcal{L}(f_{\theta_i}(x), f_{\theta^*}(x)\right]} \, d\theta_i. \tag{7}$$

Note that often we will *not* have access to the true input distribution $\mathcal{P}(x)$ to compute $\mathbb{E}_x\left[\mathcal{L}(f_{\theta_i}(x), f_{\theta^*}(x))\right]$. In that case, we can also estimate it from samples.

## 4.2 APPROXIMATING THE FRM OBJECTIVE BY LEVERAGING OVER-PARAMETERIZATION

Equation 7 is an integral in high dimensions under a non-linear constraint. In general, this is well-known to be computationally challenging. Fortunately, for this particular class of systems, we can rely on over-paramterization to propose a reasonable approximation. First, as a sanity check, we observe that all constraints $f_{\theta_i}(x_i) = y_i$ are independent and that they all have a viable solution, as we are only trying to fit each single data-point $(x_i, y_i)$ with the entire parameter set $\theta_i$. For instance, even a constant model $f(x) = c$ fits the data with $c_i = y_i \; \forall i$. In other words, the system $(\theta^*, \theta_1, \ldots, \theta_n)$ is always over-parameterized.

Moreover, it is often extremely over-parameterized. For reasonably parameterized models this is indeed the case: even small models of $10^4$ parameters (compared to modern models of more than $10^{10}$ parameters) may be underparameterized w.r.t. the entire dataset, but *extremely* over-parameterized w.r.t. fitting a single data point. Therefore, similar to the Neural Tangent Kernel literature (Jacot et al., 2018) for extremely wide neural networks, we can assume that a very small perturbation will be enough to fit each datapoint [1].

Now, we assume that we only need to analyze small perturbations $\Delta_i$ around a parameter $\theta^*$ for $|\Delta_i| \ll 1$. We can therefore take the Laplace approximation of the probability distribution we want to fit and assume it is a Gaussian with mean at $\theta^*$: $\mathcal{N}(\theta, H_{f,\mathcal{L},\theta^*}^{-1}), (H_{f,\mathcal{L},\theta^*})_{j,k} := \frac{\partial^2 \mathbb{E}_x[\mathcal{L}(f_{\theta+\Delta}(x), f_\theta)]}{\partial \Delta_j \partial \Delta_k}$. Similarly, we can take the first-order Taylor approximation of the function $f_{\theta+\Delta}(x_i) \approx f_\theta(x_i) + J_\theta f_\theta(x_i)^T \cdot \Delta$, assuming it is linear. Omitting the normalizer term, this leads to:

$$\arg\max_{\theta^*} \sum_{(x_i, y_i)} \log \int_{\Delta_i : f_\theta(x_i) + J_\theta f_\theta(x_i)^T \cdot \Delta_i = y_i} \frac{e^{-\Delta_i^T \cdot H_{f,\mathcal{L},\theta^*} \Delta_i}}{Z(\theta^*)} \, d\theta_i. \tag{8}$$

Under these conditions, computing the likelihood of $f_{\theta+\Delta_i}$ fitting $x_i$ involves integrating a gaussian distribution over either a subspace (for regression) or a half-space (for binary classification).

**Regression** We first note that the integral of the gaussian under a constraint can be seen as the pdf of $y_i \sim J_\theta f_\theta(x_i) \cdot \Delta + f_\theta(x_i), \Delta \sim \mathcal{N}(0, H_{f,\mathcal{L}}^{-1})$. Because it is a fixed linear transformation of a gaussian distribution it can also be expressed as a gaussian. In particular using the notation

---

[1]Note that this justifies that there is a large probability mass for $|\theta_i - \theta| \ll 1$, but it does not justify that this is an accurate approximation of the entire integral. However, this is a common and useful approximation.

$J_i := J_\theta f_\theta(x_i)$, we have $p(y_i) \sim \mathcal{N}\left(f_\theta(x_i), J_i^T H_{f,\mathcal{L}}^{-1} J_i\right)$. Computing its log-likelihood we obtain the following training objective where both $J_i$ and $H_{f,\mathcal{L}}$ depend on $\theta$:

$$\arg\min_{\theta^\star} \sum_{i=1}^n (y_i - f_{\theta^\star}(x_i))^T \left(J_i^T H_{f,\mathcal{L}}^{-1} J_i\right)^{-1} (y_i - f_{\theta^\star}(x_i)) + \sum_{i=1}^n \log\left(|J_i^T H_{f,\mathcal{L}}^{-1} J_i|\right) \quad (9)$$

**Classification** For binary classification the solution is similar, except that we integrate over a half-space instead of a hyper-plane. Thus, we take the gaussian ccdf (complementary cumulative distribution function) instead of the gaussian pdf. Therefore, to maximize the logprobability of a function fitting a point, we minimize the gaussian logcdf of the signed distance function to the decision boundary: $\min_\theta \sum_{i=1}^n \text{logcdf}(\Delta_i)$ where $\Delta_i := \text{sign}\left(\sigma(f_\theta(x_i))_{y_i} - \frac{1}{2}\right) \min_{\theta_i : \sigma(f_{\theta_i}(x_i))_{y_i} = \frac{1}{2}} |\theta_i - \theta|_{\Sigma_{f,\mathcal{L}}}$ is the signed distance to the decision boundary.

Note that in classification the best perturbation is *not* zero, but a very negative (i.e. opposite to the gradient) value, since this implies that the parameter $\theta$ is well within the correct classification region. This is also similar to regular ERM in binary cross-entropy, where we maximize the sigmoid, which has a very similar shape as the gaussian cdf.

For multi-class classification the integral is over an intersection of $C - 1$ half-spaces (comparing each class with the correct class $y_i$). The efficient integration in that case is still an active area of research (Gessner et al., 2020). Two potential alternatives may be practical: turning the training of an $n$-way classification into $n$ binary classifications, and linearizing the softmax of all incorrect classes jointly instead of linearizing each one independently.

### 4.3 FRM MAY DO EXPLICTLY WHAT OVER-PARAMETERIZED ERM DOES IMPLICITLY

It has been observed that neural networks often generalize despite memorizing the training dataset (Zhang et al., 2017; Poggio et al., 2017; Belkin et al., 2019; Nakkiran et al., 2021), seemingly contradicting classic understanding of generalization in ERM, which relies on controlled capacity.

FRM implicitly assigns to every datapoint $(x_i, y_i)$ its own latent model $f_{\theta_i}$ which fits it: $f_{\theta_i}(x_i) = y_i$. In this way, we can turn a model $f_\theta$ into an over-parameterized hyper-model. Although $\theta_i$ is unobserved in FGMs, the previous Taylor version of FRM becomes equivalent to this optimization:

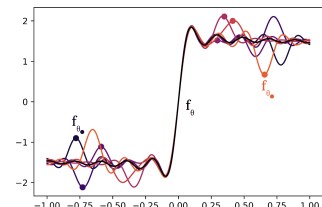

Figure 6: Minimal functional adaptations using a generalized linear model with Fourier features.

$$\min_{\substack{\theta_1, \ldots, \theta_n : \\ f_{\theta_i}(x_i) = y_i}} \sum_{i,j} |\theta_i - \theta_j|_{\mathcal{M}_{f,\mathcal{L},\theta}}^2 = \min_\theta \sum_i \min_{\substack{\theta_i : \\ f_{\theta_i}(x_i) = y_i}} |\theta_i - \theta|_{\mathcal{M}_{f,\mathcal{L},\theta}}^2 \quad (10)$$

where explicit $\theta_i$ are sought that are as close as possible according to the metric $\mathcal{M}$. Whereas ERM finds the function that best fits the data among a class of simple functions, FRM finds the simplest hyper-model to explain the data, related to the principle of Occam's Razor.

This can be seen as finding the simplest hyper-model $\{\theta_1, \ldots, \theta_n\}$ that fits the data. Simplicity is measured as the distance of parameters being close to a central parameter given a metric that captures the relationship between the function class $f_\theta$ and the loss $\mathcal{L}$. This encourages each independent function to be close to the central one, and thus all functions being close to each other, as shown in figure 6. This is related to the line of research exposed by Bartlett et al. (2021), which conjectures that ERM under gradient descent may *implicitly* find a function with two components $f_{\text{stable}} + f_{\text{spiky}}$, such that the spiky component has negligible norm but allows overfitting. In this regard, FRM can be seen as *explicilty* searching for the smallest necessary perturbation for each point.

## 5 EXPERIMENTS

To scale to neural networks, we leveraged the Taylor approximation in section 4.2. However, that requires inverting a Hessian, which is usually too big to even instantiate in memory. We bypassed this

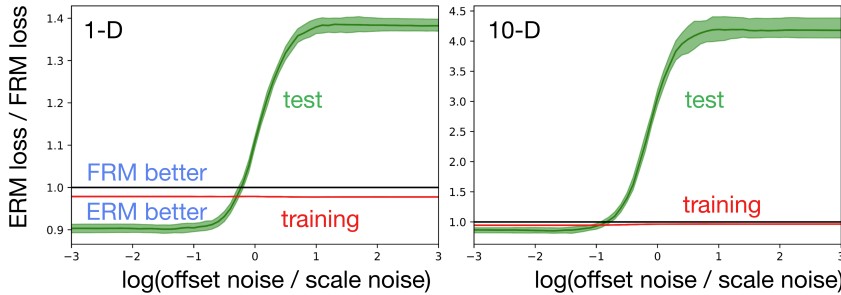

Figure 7: Ratio of train and test error between ERM and FRM as a function of the ratio between noise in the scale vs. offset components in 1-D and 10-D linear regression. As expected, we can see that ERM always has lower training loss as well as slightly lower test loss (12% lower) when its assumption (gaussian noise only on the offset) is perfectly satisfied. When noise is heteroscedastic, ERM has up to 40% higher test error. In 10 dimensions, the advantage of FRM is even starker: ERM can have 4 times more test error than FRM, despite having lower training error.

problem by 1) relying on iterative solvers to avoid the cubic cost and 2) materializing only Hessian-vector products. To do so, we use JAX (Bradbury et al., 2018) and the jaxopt package (Blondel et al., 2021), which implements implicit gradients.

## 5.1 LINEAR LEAST SQUARES

To better understand the trade-offs between FRM and ERM we analyze the simple case of linear regression under mean-squared error risk. We consider a $d$-dimensional input and a one-dimensional output. The classic ERM solution minimizes the risk on the training data: $\min_{\lambda,\beta} \sum_{(x_i,y_i)} \left(\lambda \cdot x_i + \beta - y_i\right)^2$. This is equal to doing maximum likelihood on a fixed gaussian noise on $\beta$. Thus, we expect ERM to do well in this situation, but not necessarily otherwise.

For linear regression with squared loss, the Taylor approximations in section 4.2 are exact. Furthermore, both the Hessian and the gradients are independent of the parameters, which further simplifies the objective function to just a specific re-weighting of the per-point risks: $\min_{\lambda,\beta} \sum_{(\mathbf{x}_i,y_i)} \frac{(\lambda \cdot \mathbf{x}_i + \beta - y_i)^2}{[\mathbf{x}_i,1]H^{-1}[\mathbf{x}_i,1]^T}$, with $H = \mathbb{E}_\mathbf{x}\left[[\mathbf{x},1]^T[\mathbf{x},1]\right]$. Figure 7 shows that indeed ERM does slightly better with gaussian noise in the bias, but FRM does much better when the noise is entirely in the slope. We also observe that the FRM is more than 4 times better in higher dimensions.

## 5.2 VALUE FUNCTION ESTIMATION

We demonstrate here that the proposed approach can be broadly applied on an illustrative offline value estimation task using the mountain car domain (Sutton & Barto, 2018). We consider the problem of learning a linear value function using a $15 \times 15$ grid of radial basis functions (RBFs) using the 1-step temporal difference (TD) error (Sutton, 1988) as the training loss function and using sampled transition gathered by a near-optimal policy. Both approaches were optimized with stochastic gradient descent with a constant learning rate best suited for it, selected by a grid search over hyper-parameters, and a batch size of 256. Performance is then evaluated using the root mean squared error (RMSE) between predictions and the true values on unseen samples.

We consider two different arrangements of RBFs, a uniform layout and one that is denser towards the center of the environment. Note that although the true value function has a discontinuity spiraling out from the center, which might benefit from finer resolution, the more poorly conditioned nature of this non-uniform arrangement of features makes the problem harder, as can be seen in figure 8. We see that FRM is competitive in the easier of the two cases while outperforming ERM by over 20% in the harder one. We hypothesize that TD loss is commonly subject to complex noise that can severely hinder ERM when its features are poorly aligned. Furthermore, due to the use of *bootstrapping* $(\mathcal{L}(s,r,s') := (f_\theta(s) - r - \gamma f_\theta(s'))^2)$ the temporal difference error is inherently functional through the term $f_\theta(s')$ affecting the label.

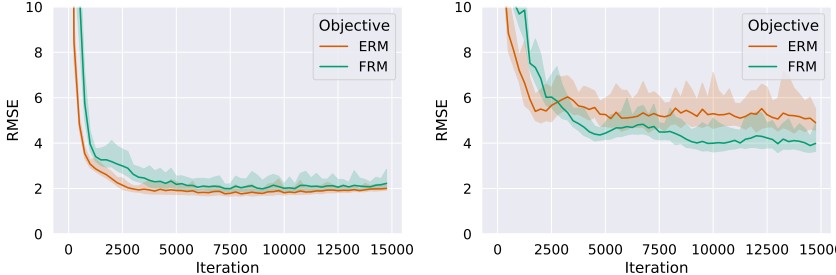

Figure 8: Comparison of the RMSE for ERM and FRM for the learned value function in mountain car under a fixed policy using a temporal difference loss with different features: **(left)** using a uniform grid of radial basis functions, **(right)** using a distorted grid of radial basis functions denser in the middle. Solid lines are the average over 20 seeds; shaded areas show the 95th percentile interval.

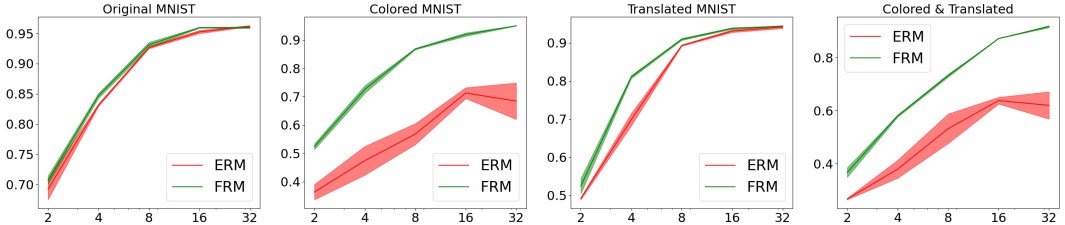

Figure 9: Accuracies of an MLP trained from latents of two CNN-based VAEs, trained with ERM and FRM. FRM provides small gains in vanilla MNIST, and large gains in all three variants.

### 5.3 FGM-BASED VAE FINDS BETTER REPRESENTATIONS WITHIN STRUCTURED VARIATIONS

To better understand when FRM works better than ERM, we build a Variational AutoEncoder(VAE) on top of MNIST (LeCun et al., 1998) and combinations of two popular variations: colored MNIST (Arjovsky et al., 2019) and translated MNIST (Jaderberg et al., 2015). We build a vanilla VAE with MLP encoder and CNN decoder. Then, we evaluate the quality of the representation to do classification for the vanilla VAE and an FGM-based decoder where noise is modeled in function space. For FGM, we train a small MLP on top of the latent representation, with a stop-gradient, and measure accuracy depending on the size of the latent. We see that in MNIST, where natural variations in orientation, translation, and color have been unnaturally removed, some gains exist but are small. In the datasets containing variations in color or translation, the FRM gains are substantial. This is because noise in CNN weights can easily explain these structured variations, as shown in figure 2. Similarly, papers such as Deep Image Prior (Ulyanov et al., 2018) have argued that neural networks are good models for real-world variability, making FRM particularly appealing for modeling real-world data. Results are shown in figure 9.

## 6 CONCLUSION

The main limitation of FRM in its current form is its compute cost. Thanks to the approximations proposed in sections 4.2 and 5 we can run FRM on a ResNet-50 using a single GPU, but with a prohibitive iteration cost. However, long term, FRM could be orders of magnitude more efficient than ERM-based approaches. As explained in section 4.3, under-parameterized FRM may behave similarly to over-parameterized ERM by making models have $n$ times more parameters $\theta_1, \ldots, \theta_n$. There, each $\theta_i$ is instantiated on the fly for loss computation and thus doesn't need to be in memory, this could provide orders of magnitude of benefit for modern datasets where often $n > 10^6$.

In the last years, there has been a clear tendency towards building large models capable of performing many tasks which were previously modeled individually. FGMs propose the natural step to model the diversity in these datasets in function space rather than output space, allowing for richer and more meaningful noise models. Despite noise being pervasive across real-world data, modern deep learning approaches still largely use simple unstructured noise models. As we keep moving towards larger, more varied datasets, properly modeling the internal data diversity will become crucial. We believe FRM provides a first step towards an effective solution.

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

Table 1: FRM outperforms ERM in a small CNN environment despite ERM having 0 training loss. Furthermore, the Hessian can be enough to express the dependence on the loss function.

| Objective | | Train | | Test | |
|---|---|---|---|---|---|
| | | positives | negatives | positives | negatives |
| ERM | positives | **.000 $\pm$ .000** | .283 $\pm$ .016 | .130 $\pm$ .006 | .278 $\pm$ .013 |
| | negatives | .336 $\pm$ .020 | **.000 $\pm$ .000** | .323 $\pm$ .018 | .119 $\pm$ .005 |
| FRM | positives | .052 $\pm$ .002 | .109 $\pm$ .007 | **.085 $\pm$ .004** | .124 $\pm$ .006 |
| | negatives | .131 $\pm$ .010 | .052 $\pm$ .002 | .136 $\pm$ .009 | **.084$\pm$.004** |

## A  FUNCTIONAL NOISE IN A CNN

To show the value of the Taylor approximation, we create a dataset by sampling different parameter assignments on a 4-layer CNN architecture. The CNN takes in a CIFAR-10 image and outputs a real number. We provide only 8 labels to each method, allowing empirical risk minimization to easily memorize the dataset. Despite FRM obtaining substantially higher training losses (.000 vs .052), we observe FRM obtains significantly less test error (.125 to .085).

We also test the ability of FRM to modify its training depending on the loss function. Although this is obviously the case for ERM, in approximate FRM the loss function enters only in an indirect way, affecting the hessian in equation 8. We modify the objective by creating two different losses, which assign zero loss to labels that are either positive or negative, respectively. Table 1 shows that indeed FRM performs better when trained and tested on the same loss (0.085 vs 0.128).

## B  PROOFS OF EMPIRICAL LOSSES BEING SUB-CASES OF FUNCTIONAL LOSSES

### B.1  MEAN-SQUARED ERROR AND $L_1$ LOSS AS A FUNCTIONAL LOSSES

Let our dataset $\mathcal{D}_{train} = \{(x_i, y_i)\}_{i=1}^n$, $y_i \in \mathbb{R}^1$, and let $\mathcal{L}_{MSE} = \frac{1}{n} \sum_{i=1}^n (f(x_i) - y_i)^2$, i.e. the mean-squared error loss.

**Lemma 1.** *For any arbitrary function class $f_{\theta,\beta}(x)$ expressible as $f_{\theta,\beta}(x) = f_\theta(x) + \beta$, there exists a functional loss restricted to functional adaptations $\theta_i = \theta$ that only change $\beta \to \beta_i$ which is equivalent to the mean-squared error loss.*

**Proof** Since we can only change $\beta$ there is a single solution to the per-point constraint: $f_\theta(x_i) = f_\theta(x_i) + \beta_i = y_i \Rightarrow \beta_i = y_i - f_\theta(x_i)$. We can now model the probability distribution over functions $\mathcal{F}(\theta, \beta_i | \theta, \beta, \mathcal{L}_{MSE})$ as a gaussian centered at $(\theta, \beta)$. Since $\theta$ doesn't change, this will just be $\mathcal{N}(\beta_i - \beta)$. Maximizing the mean of the log-probabilities will result in $\frac{1}{n} \sum_i \log \mathcal{N}(\beta_i - \beta) = \frac{1}{n} \sum_i (\beta_i - \beta)^2 = \frac{1}{n} \sum_i (y_i - f_\theta(x_i) - \beta)^2 = \frac{1}{n} \sum_i (y_i - f_{\theta,\beta}(x_i))^2 = \mathcal{L}_{MSE}$.

Of note, the Gaussian model of the functional distribution satisfies

$$\mathcal{F}(\theta, \beta_i | \theta, \beta, \mathcal{L}_{MSE}) = \mathcal{N}((\theta, \beta_i) - (\theta, \beta)) \propto e^{-|\beta - \beta_i|^2} = e^{-\mathbb{E}_x \mathcal{L}_{MSE}(f_{\theta,\beta}, f_{\theta',\beta'})}.$$

This is because for all $x$, $\mathcal{L}_{MSE}(f_{\theta,\beta}(x) - f_{\theta',\beta'}(x)) = |f_{\theta,\beta}(x) - f_{\theta',\beta'}(x)|^2 = |\beta - \beta'|^2$.

Finally, we note that the entire derivation can be equivalently followed for the L1 loss by swapping $|\cdot|^2$ for $|\cdot|$ and the Gaussian distribution for the Laplace distribution.

### B.2  CLASSIFICATION ERROR AS A FUNCTIONAL LOSS

Let us now look at multi-class classification and let our dataset $\mathcal{D}_{train} = \{(x_i, y_i)\}_{i=1}^n$, $y_i \in \{1, \ldots, C\}$. Our function class will output in an unconstrained *logit* space $\mathbb{R}^C$ and we define $\mathcal{L}_{cls} = \frac{1}{n} \sum_{i=1}^n \mathbb{1}[\![y_i = \arg\max_c (f_{\theta,\beta}(x_i))_c]\!]$, i.e. the classification error. As in previous sections, abusing notation we will refer to $\mathbb{1}[\![y_i = \arg\max_c (f_{\theta,\beta}(x_i))_c]\!]$ as $\mathbb{1}[\![y_i = f_{\theta,\beta}(x_i)]\!]$.

**Lemma 2.** *For any arbitrary function class $f_{\theta,\beta}(x)$ expressible as $f_{\theta,\beta}(x) = f_\theta(x) + \beta$, $\beta \in \mathbb{R}^c$, constrained on $f_\theta(x)$ being finite, there exists a functional loss restricted to functional adaptations $\theta_i = \theta$ that only change $\beta \to \beta_i$ which is equivalent to the classification error.*

**Proof** We will show that a solution is given by $\mathcal{F}(\theta, \beta_i | \theta, \beta, \mathcal{L}_{cls}) = p \cdot \delta(\beta_i - \beta) + (1 - p) \lim_{\sigma \to \infty} \mathcal{N}(0, \sigma)(\beta)$, with $p = \frac{e-1}{C+e-1} \in (0, 1)$. In other words, a specific positive (note the open brackets) combination of an infinitely-sharp distribution (Dirac's delta) with an infinitely-flat distribution. Given a fixed $p, \theta, \beta$, the probability of $y_i = \arg\max_c f_{\theta_i, \beta_i}(x_i)$ will be equal to $p \cdot \left[y_i = \arg\max_c \left(f_{\theta,\beta}\right)_c\right] + \frac{1-p}{C}$. This comes directly from the definition of the functional probability distribution: with probability $p$, we have $(\theta_i, \beta_i) = (\theta, \beta)$ and thus the result depends solely on $(\theta, \beta)$; with probability $(1 - p)$ the logits are perturbed by an infinitely strong noise and thus the $\arg\max$ will just be a uniform distribution over the classes, i.e. $\frac{1}{C}$.

Now, the average log-likelihood of the functional loss will be:

$$\frac{1}{n}\sum_{i=1}^n \log\left(p \cdot \mathbb{1}[\![y_i = f_{\theta,\beta}(x_i)]\!] + \frac{1-p}{C}\right) =$$

$$\log\frac{1-p}{C} + \frac{1}{n}\sum_{i=1}^n \log\left(\frac{p \cdot \mathbb{1}[\![y_i = f_{\theta,\beta}(x_i)]\!] + (1-p)/C}{(1-p)/C}\right) =$$

$$\log\frac{1-p}{C} + \log\left(\frac{p + (1-p)/C}{(1-p)/C}\right)\frac{1}{n}\sum_{i=1}^n \mathbb{1}[\![y_i = f_{\theta,\beta}(x_i)]\!] =$$

$$\log\frac{1-p}{C} + \log\left(1 + \frac{pC}{1-p}\right)\mathcal{L}_{cls} =$$

$$-\log\left(C + e - 1\right) + \mathcal{L}_{cls}.$$

where in the second step we observe that the log term within the sum is zero when $y_i \neq f_{\theta,\beta}(x_i)$ and, in the last step, we have set $p = \frac{e-1}{C+e-1}$, which by construction is in $(0, 1)$. We can now easily see that this is equivalent to $\mathcal{L}_{cls}$ up to a constant additive term, which will not affect any optimization.

### B.3 Cross-entropy loss as a functional loss

Continuing in multi-class classification and let our dataset $\mathcal{D}_{train} = \{(x_i, y_i)\}_{i=1}^n, y_i \in \{1, \ldots, C\}$. Our function class will output in an unconstrained *logit* space $\mathbb{R}^C$ and we define $\mathcal{L}_{CE} = \frac{1}{n}\sum_{i=1}^n \log\sigma\left(f_{\theta,\beta}\right)_{y_i}$, i.e. the cross-entropy loss. Here, $\sigma(\cdot)_c$ corresponds to taking the $c$-th component of the softmax of a given logit to obtain the probability of a given class $c$ given the logit predictions.

**Lemma 3.** *For any arbitrary function class $f_{\theta,\beta}(x)$ expressible as $f_{\theta,\beta}(x) = f_\theta(x) + \beta$, $\beta \in \mathbb{R}^C$, there exists a functional loss restricted to functional adaptations $\theta_i = \theta$ that only change $\beta \to \beta_i$ which is equivalent to the cross-entropy loss.*

**Proof** As shown in (Jang et al., 2016; Maddison et al., 2016) if we have logits $\gamma_c = f_\theta(x_i)_c + \beta_c$ we can sample from the probability distribution of distribution equal to $\sigma(\gamma)$ by $c = \arg\max_i(\gamma_i + g_i)$ where each $g_i$ follows an independent Gumbel distribution, i.e. $g_i = -\log(-\log u_i), u_i \sim \mathcal{U}(0, 1)$. This gives us a trivial expression for a functional distribution over which to make maximum likelihood: $\beta_i \sim \beta + \mathcal{G}$, where $\mathcal{G}$ consists of $c$ independent Gumbel noise variables. This is because, since $\beta$ lives in logit space, adding noise to $\beta$ is equivalent to adding noise to the logits. Finally, since the cross-entropy loss is the maximum likelihood assuming a probability distribution given by the logits and we have shown a functional distribution with the same distribution, performing maximum likelihood on that distribution is equivalent to minimizing the cross-entropy loss.

## C Universal Distribution Theorem

**Definition 2.** Given a function class $\mathcal{F}$ with parameterisation $\Theta$, we define a Functional generative model $(P(x), P(\theta)) \in FGM[\mathcal{F}_\Theta, \mathcal{X}]$ as a probability density function $p(x, y) \in L^2[\mathcal{X} \times \mathcal{Y}]$ with $x \sim P(x) \in L^2[\mathcal{X}]$, and $y = f_\theta(x), \theta \sim P(\theta) \in L^2[\Theta]$.

Note that, in particular, $P(\theta \in \Theta)$ and $P(x \in \mathcal{X})$ are independent and $y$ is deterministic given $x, \theta$; as shown in figure 1.

**Theorem 2** (**Universal Distribution Theorem**). *Let $q(x, y) \in L^2[\mathcal{X} \times \mathcal{Y}], \mathcal{X} = [0, 1]^n \subset \mathbb{R}^n, \mathcal{Y} = [0, 1]^m \subset \mathbb{R}^m$ be a given probability density distribution function. Let $\mathcal{F}_\Theta^k$ be the class of 3-layer neural networks with sigmoidal activation function and $k$ neurons in the hidden layer. For any $\epsilon > 0$, $\exists K$ and a functional generative model $(P(x), P(\theta)) \in FGM\left[\mathcal{F}_\Theta^K, \mathcal{X}\right]$ s.t. $D_{TV}\left((P(x), P(\theta)), q\right) < \epsilon$, with $D_{TV}$ being the total variation distance.*

For the first layer we use deterministic weights with arbitrarily-big slope to implement the functions $1[\![x_i \geq c_j]\!]$ for all coordinates $1 \leq i \leq n$ and $c_j = \{-1, -1 + \epsilon, \dots, 1 - \epsilon, 1\}$. For the second layer, we again use deterministic weights to implement functions $1[\![\mathbf{x} \in [a_1, a_1 + \epsilon) \times \cdots \times [a_n, a_n + \epsilon)]\!]$ to determine whether a given input is within a hyper-cube of side $\epsilon$. Exactly one of those two-layer nodes will be active for any given input. From the node corresponding to $[a_1, a_1 + \epsilon) \times \cdots \times [a_n, a_n + \epsilon)$ the each of the output nodes there are $m$ weights $\theta$, we assign them a distribution equal to $\theta_{1:m} \sim P(y | x = (a_1, \dots, a_n))$. Because $P(y|x)$ is continuous, $P(y|x = (a_1, \dots, a_n))$ will be arbitrarily close to $P(y|x)$ for any $x$ in the hyper-cube $[a_1, a_1 + \epsilon) \times \cdots \times [a_n, a_n + \epsilon)$ for a sufficiently-small $\epsilon$.

We note that this universality also holds for a 2-layer neural network as well (also a universal function class). However, the prove for that case is more cumbersome and less insightful for our purposes.

# D FURTHER UNDERSTANDING THE DIFFERENCE BETWEEN ERM AND FRM

**The ERM assumption:** by assuming that the training objective is equal to the test loss $\mathcal{L}$, ERM can be suboptimal for certain $\mathcal{P}(\theta)$, like the house example on section 3. As shown in appendix B, for many loss functions $\mathcal{L}$, including most of the common ones, ERM is equivalent to assuming the functional generative model and then doing maximum likelihood on $\mathcal{P}(\theta)$ by assuming it has a form parameterized by $\hat{\theta}$ whose uncertainty is only on the output offset parameters. In other words, the assumption equivalent to performing ERM is often strictly more assuming than FGMs.

For instance, consider predicting the price of different houses as a function of their size and having MSE as the loss. Doing empirical risk minimization with the MSE would be equivalent to doing maximum likelihood on the following price model: $y_i \sim \mathcal{N}(f(x_i), \sigma^2)$. However, we would expect noise to be *heteroskedastic* with higher variations for higher prices. Thus, even if we are evaluated on MSE on the test data, it may not be advisable to use it as our training criteria.

Similarly, consider a child learning a concept from examples on a textbook rather than from standardized images of a dataset. Images may receive different illuminations from the sunlight, or be in different positions than we expect. These factors will produce massive changes in pixel space, but in very structured ways (fig 2). However, humans can still easily grasp the idea because the 'conceptual' noise is small.

How can we have more meaningful noise models? By construction, we will often believe that the function class $f_\theta$ is a good characterization of the relationship between $x$ and $y$. It is thus a natural assumption to define a noise model by leveraging the function class itself. More concretely, we can think of a generative model of the data as first sampling the input $x_i$, then sampling a function $f_i \sim \mathcal{F}(\mathcal{L}, \theta)$ from some parameterized distribution over functions, which will depend on both the problem-specified loss function $\mathcal{L}$ as well as the function class $f_\theta$. Once the function and the input have been sampled, the output is automatically determined $y_i = f_i(x_i)$, see the right of figure 1.

For example, in our house-price prediction, if we are using a linear model $f(x) = \lambda \cdot x + \beta$, then it makes sense to think about our data as coming from first sampling $x_i \sim p(x)$ and $(\lambda_i, \beta_i) \sim \mathcal{F}(\mathcal{L}, (\lambda, \beta))$, then computing $y_i = \lambda_i \cdot x_i + \beta_i$, as shown on the right of figure 1. For instance, $\beta_i$ can model different commissions or taxes, and $\lambda_i$ can model the per-meter-square price being variable across neighborhoods. Even if we care about making accurate predictions in dollar-space, assuming our uncertainty is only in the offset term $\beta_i$ may be too restrictive.

### D.1 ERM VS FRM FOR THE LINEAR CASE

Let us now take a deeper look at our linear regression example. We have a dataset $\mathcal{D}_{train} = \{(x_i, y_i)\}$, depicted in the top-right of figure 3a, with an arbitrary color per point. For every point, there is a subspace of models $(\lambda_i, \beta_i)$ s.t. $\lambda_i x_i + \beta_i = y_i$. Since we only have two parameters, we can also look at function-space in 2-D, and plot the corresponding subspace for each point, in the bottom-left of figure 3a. We observe that every point gives us a line in function space, which we plot with the corresponding color.

Our goal is to produce a probability distribution $\mathcal{P}(\lambda, \beta)$ such that the sum of the log-densities of each line $(\lambda_i, \beta_i)_{\lambda_i x_i + \beta_i = y_i}$ is maximal. Intuitively, this means that each line should pass through a high-density area of the probability distribution, but it does not mean that the line should be covered by the high-density area (which is not possible, since they're unbounded). This can be seen in figure 3b where all lines pass near the center of the distribution generating the data (marked in green).

We can further see that ERM with the MSE loss is equivalent to finding a point $(\lambda^{ERM}, \beta^{ERM})$ that minimizes the *vertical* distance to each line:

$$
\begin{aligned}
(\lambda^{ERM}, \beta^{ERM}) &= \min_{\lambda, \beta} \sum_i (y_i - \lambda x_i - \beta)^2 \\
&= \min_{\substack{\lambda, \beta, \\ \lambda^i : \lambda^i = \lambda}} \sum_i (y_i - \lambda_i x_i - \beta)^2 \\
&= \min_{\substack{\lambda, \beta, \{\lambda^i, \beta^i\} : \\ \lambda^i = \lambda, \\ \lambda_i x_i + \beta_i = y_i,}} \sum_i (\beta_i - \beta)^2.
\end{aligned}
$$

In contrast, if the probability distribution in parameter space is a Gaussian, FRM involves taking the distance of the entire vector $(\lambda, \beta)$, using the inverse covariance matrix as the metric. For cases where most of the uncertainty is in the slope, as in figure 3b, ERM measures the distance in the vertical direction and FRM measures it almost horizontally, leading to different results.

### D.2 VISUALIZATION FOR A SIMPLE FULLY CONVOLUTIONAL NETWORK

Figure 2 shows the difference between MSE and its functional correspondent for a small fully-convolutional network mapping images to images $f_\theta$. Images $y$ with the same empirical loss $|y - f_\theta(x)|^2$ could require very different functional adaptations to explain: $\min_{\theta' : f_{\theta'}(x) = y} |\theta' - \theta|_{f, \mathcal{L}}$. For instance, if one does edge detection and mistakenly translates its prediction a bit to the right, this small change in functional space could lead to a large error in pixel space. Similarly, if we have a pattern detector and we slightly change its threshold, it could make the entire prediction darker or lighter.

Conversely, if we add unstructured noise onto our image, it is to be expected that it will have a high functional loss as no small perturbation of the function could simultaneously explain pure noise. That's indeed what we observe in figure 2b when we look for images with high and low functional loss for a fixed empirical loss. Images with high functional loss contain salt-and-pepper-like noise that breaks the smooth pattern of the original image. In contrast, images with low functional loss preserve the overall structure while uniformly shifting large blocks of pixels to a much lighter color. If the noise in our data is better represented by our functional class than noise in the output, we can take this into account to improve learning.

