# OpenReview forum: "Functional Risk Minimization"
_ICLR.cc/2023/Conference — Submitted to ICLR 2023_

### Official Review · Reviewer_EeLk · 2022-10-25

**Confidence:** 4
**Correctness:** 3
**Technical Novelty And Significance:** 4
**Empirical Novelty And Significance:** 3
**Recommendation:** 6

**Clarity, Quality, Novelty And Reproducibility:**

It is well-written article with high quality and novelty. I do not check the reproducibility of this paper.

**Strength And Weaknesses:**

Pro: The paper rethink the common assumption in machine learning community and proposes a new comprehensive model that can subsumes classic Emipirical Risk Minimization for many common loss functions.
Cons: 1) The computation cost of FRM in its current form is high. 2) Maybe the authors can discuss more why the FRM is really crucial and why we could not just modify the classic ERM for instance and get similar performance.

**Summary Of The Paper:**

This paper proposes a general framework Functional Risk Minimization (FRM), a general framework for scalable training objectives which results in better performance in small experiments in regression and reinforcement learning. FRM model each data point (x_i, y_i) as coming from its own function f_\theta_i. The authors also show that FRM can be regraded as finding the simplest model that memorizes the training data, providing an avenue towards understanding generalization in the over-parameterized regime.



**Summary Of The Review:**

The paper is marginally above the acceptance line. I would suggest to accept this paper.

---

> ### Author Response · Authors · 2022-11-19
> **Thanks for your review. Comments on cost and why it's worth building a completely novel framework**
>
> Thanks for your detailed and positive review!
> Many of your comments and questions are addressed in the global response, where we clarify why FRM is scalable and show new experiments in unsupervised representation learning using FRM and CNNs.
> >1) The computation cost of FRM in its current form is high.
>
> We show experimentally that thanks to the approximation in section 4.2, FRM scales to CNNs. Although slower than ERM, we explain how and why FRM could scale to larger experiments. This is detailed in the global response.
>
> > 2) Maybe the authors can discuss more why the FRM is really crucial and why we could not just modify the classic ERM for instance and get similar performance.
> There are two questions in this comment:
>
> ***When is FRM more useful?*** The short answer is when variations are structured, as often happens in real data. More details, particularly on how this is supported by the new experiments in the PDF, in the global response.
>
> ***Why do we need a drastic change instead of an incremental change on ERM?*** We see this paper as a first step towards changing one of the foundations of current ML. As a first step, experiments are small scale (but now include proper neural networks and popular benchmarks). There are three reasons for attempting a drastic change:
> - ***Because there are signs that something is fundamentally flawed with our classic understanding ERM***: the summary is that ERM was designed for the under-parameterized regime where models are too small to perfectly fit the training data. However, many works have observed that neural nets can memorize the data [1] and yet still generalize very well [2, 3]. FRM fundamentally lives in the over-parameterized regime (sec 4.3), and has relations to some of the explanations of over-parameterized neural networks. This makes us think that it can A) give us insights on over-parameterized networks and B) provide an alternative framework for modern deep learning.
> - ***Because we know the potential upshot is very big***: it is well-known that neural networks are often much larger than they need to be [4]; i.e. they contain sub-networks hundreds of times smaller that perform almost equally well or better. It is thus reasonable to think that there exist opportunities for disrupting the ERM ecosystem and get order-of-magnitude gains.
> - ***Because ERM is ubiquitous and FRM similarly has a huge range of applicability***: most ML applications rely on variants of ERM. Similarly, the assumption behind FRM is very general (and often, even more than ERM's, see sec 3.2). For instance, our experiments now apply to supervised, unsupervised, and reinforcement learning. Therefore, any progress on FRM could have a widespread impact.
>
> This paper is only the first step towards a big goal, but we have many reasons to believe it’s a path worth exploring.
>
> [1] Zhang et al. Understanding deep learning requires rethinking generalization
>
> [2] Bartlett et al. Deep learning: a statistical viewpoint.
>
> [3] Belkin et al. Reconciling modern machine-learning practice and the classical bias–variance trade-off (Double descent)
>
> [4] Frankle et al. The Lottery Ticket Hypothesis

---

### Official Review · Reviewer_6yNo · 2022-10-25

**Confidence:** 3
**Correctness:** 3
**Technical Novelty And Significance:** 3
**Empirical Novelty And Significance:** 3
**Recommendation:** 5

**Clarity, Quality, Novelty And Reproducibility:**

Overall, the paper is clear and the idea is interesting, but there are some parts need more discussions or explanations.

**Strength And Weaknesses:**

Strength

1. the idea of associating each data point with a function is interesting.
2. the paper is well-written and examples are given to show the motivation and clear some concepts.
3. the functional risk minimization may extend the existing ERM to a new era.

Weakness

1.  Even the authors claims there are some difference between the proposed FRM to the Bayesian learning on page 2, it is still confusing to me that the proposed model, at least the functional generative model, is just a kind of hierarchal Bayesian model as illustrated in Fig 1. The authors claim that 'FRM only use a single parameter at test-time'. It looks weird because FRM believes every data point is associated with its own function f_{\theta_i} then why and how you use single \theta at test time? like generate new data?

2. The similar thing happens to FSM for supervised learning. When training using (1) and predicting using single \theta^*, the objective function for training and test will be different which implies that the assumptions behind training and test are different. Is that OK? Note that this is not happened in ERM where the assumptions behind training and test are consistent. Is there any effect from such property?

3. There is no experimental evaluation on the proposed functional generative model in section 5.




**Summary Of The Paper:**

This paper proposes a new basic data generation framework where each data point is associated with a function and the `noises' are assumed to be from the variation of its function rather than its output used in classical ERM.  Two corresponding models are proposed for unsupervised learning (generative model) and supervised learning.

**Summary Of The Review:**

This paper proposes an interesting and valuable idea to extend the classical ERM to FRM. The benefits are well introduced. However, there are sill some confusions of the concepts, like the relation with hierarchal Bayesian. Please see Strength and Weakness for more details.

---

> ### Author Response · Authors · 2022-11-19
> **Thanks for your review. Clarifying $\theta_i$ vs $\theta^*$ and new experiment**
>
>
> Thanks for your review and the kind words on FRM extending ERM to a new era.
> > Even the authors claims there are some difference between the proposed FRM to the Bayesian learning on page 2, it is still confusing to me that the proposed model, at least the functional generative model, is just a kind of hierarchal Bayesian model as illustrated in Fig 1.
>
> This is a good question, we’ll improve the writing to make it clearer. Bayesians put a prior on the parameters, but assume there is a single “true” parameter per dataset. They then explain the noise via some other mechanism in output space. Hierarchical Bayes models are used when we have multiple datasets, such as in multi-task learning.
>
> Under this Hierarchical Bayes view, we are taking the single-task setting and assuming that every point is its own dataset, but that there is no noise. Therefore, everything is now explained in function space.
>
> A Bayesian would say: “With finite data, I don’t know which parameter describes this dataset, but I know it’s a single one. It cannot perfectly explain every point though, because there is output noise.”
>
> We would say: “No single parameter describes all points in the dataset. Each point has its own. However, each per-point parameter perfectly explains its point, no further noise is required.”
>
> This leads us to your follow-up question.
>
> >  The authors claim that 'FRM only use a single parameter at test-time'. It looks weird because FRM believes every data point is associated with its own function $f_{\theta_i}$ then why and how you use single $\theta$ at test time? like generate new data?
>
> As you say, every point has its own $f_{\theta_i}$, just as each point has its own $y_i$. We have the same goal as ERM: come up with a single $\theta$ that does well for unseen samples. Whereas at training time we can see both $x_i$ and $y_i$, at test time we only see $x_i$, so we have no hope to get a perfect $\theta_i$. Therefore, at test-time we simply predict $f_\theta(x_i)$ and “hope for the best” (of course, backed up by our assumptions and our math), as done in ERM-based ML at test time.
>
> In other words, FGMs are a modeling assumption, akin to assuming that our data is translation invariant: we consider the existence of $f_{\theta_i}$ to derive our training objectives, and from this training procedure we obtain some $\theta^*$. At test-time, we just use $\theta^*$ like ERM does.
>
> > The similar thing happens to FSM for supervised learning. When training using (1) and predicting using single $\theta^*$, the objective function for training and test will be different which implies that the assumptions behind training and test are different. Is that OK?
>
> Related to our explanation above, we do not believe any $\theta^*$ will explain our test data perfectly. This would require $f_{\theta^*}$ having 0 test loss. The goal of $\theta^*$ is to result in low test loss. Therefore, $\theta^*$ is the best parameter $\theta$ to use at test-time given our inference of $p(\theta)$ from training-time. We will clarify this in the writing.
>
> > There is no experimental evaluation on the proposed functional generative model in section 5.
>
> Now there is :) see the general response. In particular, our FRM-trained CNN decoder is similar to that used to generate figure 2. Thanks for the suggestion!

---

### Official Review · Reviewer_vBxE · 2022-11-03

**Confidence:** 3
**Correctness:** 2
**Technical Novelty And Significance:** 4
**Empirical Novelty And Significance:** 3
**Recommendation:** 3

**Clarity, Quality, Novelty And Reproducibility:**

Clarity is poor, as explained above.

The novelty and potential significance are high.

The code is provided in the supplement for reproducibility purposes, but I have not checked it.

**Strength And Weaknesses:**

### Strengths

1. This paper presents a very interesting and (to my knowledge) novel take on representing noise in a learning framework. This could be highly impactful as it may allow for more easily dealing with the variation in real datasets than standard MLE in additive noise models.

2. The empirical results on linear regression with non-uniform noise seem to be strong, demonstrating that as the noise distribution changes, FRM begins to outperform ERM.

### Weaknesses

1. It is not clear how closely the implementable algorithm relates to the theoretical presentation and derivations. Equations 7 and 8 seem to be the core algorithmic contribution of the paper, but they are never formally derived and it is highly unclear under what conditions they will actually relate to the population objective.

2. The implementable algorithm as proposed does not seem to be scalable. In particular, it requires both a significant approximation to integrate over parameters and then uses Hessian information that could be difficult to get for large models. This is born out by the fact that all the experiments, while interesting, are quite small in scale.

3. There is no formal argument made as to why or when FRM will outperform ERM. In fact, there are no arguments at all showing that (a) the FRM minimizer will achieve low loss, (b) the finite-sample version of FRM approximates the population variant, or (c) the finite-sample FRM minimizer will achieve low loss.

4. The presentation is confusing.

    a. The use of "ERM" is non-standard and confusing. ERM, as used in foundational work like [1], simply refers to minimizing an empirical loss as a proxy for an expected loss. There is no mention of any particular assumption about additive noise in the ERM principle. Moreover, the entire reason to use the ERM term is to contrast it with learning objectives like structural risk minimization. This paper instead uses ERM to refer to the noise model induced by ERM with particular loss functions by viewing them as MLE. It would perhaps be more clear to frame the novelty of the approach as replacing MLE under an additive noise model with a novel MLE under the functional latent variable noise model.

    b. Similarly, FRM is often used to refer to the noise model, which is in fact the FGM under the nomenclature introduced in the paper. The paper would be much improved by being more clear about the difference between the FGM modeling assumption and the FRM learning objective.

    c. The FRM learning objective is never clearly and explicitly layed out. When the authors say FRM, what implementable algorithm are they explicitly referring to? This needs to be more clear.


[1] Vapnik, V. (1991). Principles of risk minimization for learning theory. Advances in neural information processing systems, 4.


Minor: There are a few typos throughout the paper. For example, in the abstract "regraded" should be "regarded". In the "FGMs encode..." paragraph on page 4, many spaces are missing after periods. In the Table 1 caption "Furthermoe" should be "furthermore".

**Summary Of The Paper:**

This paper presents a novel framework for supervised learning problems. First, it introduces functional generative models which represent $ P(x,y)$ in terms of a latent variable $ \theta$ that is sampled independently of $ x $ and then determines $ y $ as a function $ f_\theta(x)$. This is contrasted to a standard additive noise model, where $ y = f_\theta(x) + \epsilon$. Using this noise model, the paper proposes functional risk minimization (FRM) as an objective to find the maximum likelihood parameter $ \theta^*$ and proposes an approximate algorithm that relies on approximating an integral over all parameters by a local Laplace approximation.

**Summary Of The Review:**

I think this paper clearly has some interesting and promising ideas. However, as currently presented I cannot support acceptance because of the significant issues connecting the actual algorithm to the motivation, lack of basic theory, and general lack of clarity.

---

> ### Author Response · Authors · 2022-11-19
> **Thanks for your review. Clarifying scalability, proofs, and treatment of ERM**
>
> Thank you for the detailed review and your praise for the novelty and potential impact of our work!
>
> Note that the general review addresses many of your concerns with newer experiments and clarifications on scalability. We address your other concerns here.
>
> > [Weakness 1] Equations 7 and 8 seem to be the core algorithmic contribution of the paper, but they are never formally derived
>
> Up to equation 6, things are strictly derived from our low test loss goal and our FGM assumption. With one assumption, equations 6 and 7 are also connected. In the text there is a big gap between 6&7 because we want to give intuition on eq 6. We’ve rewritten that section to make the connection clearer.
>
> Eq 8 is derived from eq 7 plus the Laplace&Taylor approximations, substituting every component for its approximation.
>
> > [W2] [...] uses Hessian information that could be difficult to get for large models. [...]
>
> We do not need to instantiate the Hessian: we only need $H^{-1}\cdot v$ for some v, so we solve the system iteratively with Hessian-vector products, each costing a couple of forward-backward passes ($O(|W|)$, not $O(|W|^2)$).
>
> The other parts of W2 are addressed in the global response, both with clarifications and experiments.
>
> > [W3] There is no formal argument made as to why or when FRM will outperform ERM. In fact, there are no arguments at all showing that (a) the FRM minimizer will achieve low loss, (b) the finite-sample version of FRM approximates the population variant, or (c) the finite-sample FRM minimizer will achieve low loss.
>
> - We believe the new experiments, together with the previous ones, show a clear story that when variations are structured (as is often the case in the real world) FRM tends to perform better than ERM if the functional class captures this structure.
> - We formally prove that FGMs subsume ERM for 4 of the most common loss functions (as you correctly say, not for all losses!).
> - We minimize an empirical sample of the functional loss(Eq 7), so we expect to have low expected functional loss. As you point out, what we want is low test loss. Now, following Eqs 1-7, when our assumptions are met, minimizing functional loss (Eq 7) implies minimizing Eq 1: expected (i.e. test) loss.
> - If the data doesn’t come from the corresponding FGM, we cannot guarantee anything.
>
> We will include these clarifications in the final paper.
>
> > ERM, as used in foundational work like [1], simply refers to minimizing an empirical loss as a proxy for an expected loss. There is no mention of any particular assumption about additive noise in the ERM principle.
>
> You are right. We were aware of that, and were careful to mention that in multiple places, but probably not enough, and we’ll add more clarifications. For instance, figure 1 is more general than an additive noise model, as it merely imposes the factorization of the likelihood using the model’s prediction, without necessarily making the final distribution additive. However, as you rightly point out, even figure 1 doesn’t describe all instances of ERM. This was worded in the caption, but not highlighted enough, and we’re improving it.
>
> At the same time, we want to point out that a large fraction of practitioners use ERM with a loss that satisfies fig 1 and even the additive noise view, such as using MSE, L1, cross-entropy, or accuracy metrics. This simplification makes the comparison to FRM much easier, minimizing complexity. Furthermore, part of our point is that ERM conflates MLE with risk minimization, and these are two separate things. FRM considers them separately, but merges them to get a unique advantage.
>
> Having said all that, we could have been clearer and we’ll improve the writing accordingly.
>
> > The FRM learning objective is never clearly and explicitly laid out.
> We do lay it out, but you’re right that we did not do it clearly enough. There are two key equations: the exact FRM equation (eq 7) and the approximation for models with many parameters (which we’ve now labeled as eq 9). We’ve added green highlight boxes around those two equations.
>
> We always use equation 9 in our experiments. For linear functions with MSE (the first two experiments in the new PDF) the approximation is exact, so we are also using equation 7.
>
> Thanks again for your review. It has greatly helped us improve the paper.

---

### Author Response · Authors · 2022-11-19
**New experiments and clarifications on scalability**

Thanks all for your thoughtful reviews. There is an agreement that the paper is novel and interesting, which we find very positive. Next, we address common concerns here and individual concerns separately.
All of you wondered about the scalability of FRM to real neural networks and about when FRM is most useful. Reviewer 6yNo also asked for experiments based on figure 2. We have designed a series of experiments to address these key concerns.

# New series of experiments
We compare ERM and FRM representations from a CNN Variational AutoEncoder(VAE) on 4 popular variants of MNIST (MNIST[1], colored MNIST [2], translated MNIST [3], and colored & translated MNIST). The ERM variant is a vanilla VAE, with noise in pixel space. The FRM variant uses noise in the CNN decoder weights to capture structured variations(see fig 2). We hypothesize that this results in better representations in the bottleneck layer of the VAE.
This is indeed the case: when we train a small MLP on top of the VAE encodings, FRM results in small gains in vanilla MNIST, and large gains in the 3 popular variants (new fig 9).
We believe this series of experiments ***addresses all three key concerns***:
- ***It shows FRM is scalable*** to real datasets and neural networks like VAEs.
- ***It offers results for image-based generative models***, as requested by reviewer 6yNo.
- ***It illustrates that FRM is more beneficial when data variations are structured***, capturing them as noise inside the decoder CNN. Similarly, it is known[11] that NNs are good models for real world variability(highly structured), making FRM particularly appealing.

# Clarifications about scalability
All of you mentioned concerns about the scalability of FRM. We address these concerns in the PDF and here. We discuss three main points:

### 1. The pure version of FRM (eq. 7) is indeed not scalable
FRM is a very general framework taking any function class. We expect customizations to particular function classes to speed up computations. In this paper, we do so for the linear case and models with many parameters.

### 2. The approximation (eq. 9) is both accurate and scalable
- ***Accurate***: Intuitively, FRM is based on 1-point datasets, for which most NNs are comparatively huge. Thus, our Taylor approximation is analog to a well-explored and productive line of work analyzing very large NNs, leading to model innovations[4-6].
- ***Scalable***: section 4.2 provides an implementation that just requires inverting a Hessian and uses recent literature to do this inversion very efficiently. Many papers leveraged similar computations[7,8] and further speeding them is an active area[9,10].
- With some reasonable simplifications, we can ***run ImageNet on a V100 with FRM, but slowly***. As with NeRF and diffusion, ***first research works are often slow***, but can often be sped up with more research and compute. We plan to work in this direction.

### 3. FRM may lead to radically increasing the scalability of NN training.
- It is well-known that NNs are much larger than they need to be[12].
- As briefly explained in section 4.3, FRM can be seen as implicitly representing a model N times larger than the base model it trains, where N is the number of datapoints (often thousands or millions).
- FRM connects to Bartlett et al.’s hypothesis on the benign overfitting of over-parameterized NNs under ERM, but FRM doesn’t require over-parameterized models. As such, ***FRM-related ideas may enable networks to beat ERM-trained networks that are hundreds of times larger***.
### Conclusion
We are aware that the paper opens more questions than it closes; as we're starting a new research direction. As reviewers mentioned, FRM is novel and interesting. Furthermore, ***our experiments now cover all three major ML variants: supervised, unsupervised, and reinforcement learning***. ERM has underpinned most of ML for decades, yet its foundations are being put in question by modern deep learning[14]. FRM offers a general alternative with many potential upsides. Therefore, ***we see this paper, not as the final piece, but rather as a big first step towards a worthy goal***.
### References
[1] LeCun et al. Gradient-based learning applied to document recognition

[2] Arjovsky et al. Invariant Risk Minimization

[3] Jaderberg et al. Spatial Transformer Networks

[4] Jacot et al. Neural Tangent Kernel

[5] Zhang et al. Fixup initialization

[6] Arora et al. Harnessing the power of infinitely wide deep nets on small-data tasks

[7] Bai et al. Multiscale deep equilibrium models

[8] Lorraine et al. Optimizing millions of hyperparameters by implicit differentiation

[9] Geng et al. On training implicit models

[10] Bai et al. Stabilizing Equilibrium Models by Jacobian Regularization

[11] Ulyanov et al. Deep Image Prior

[12] Frankle et al. The Lottery Ticket Hypothesis

[13] Bartlett et al. Deep learning: a statistical viewpoint

[14] Zhang et al. Understanding deep learning requires rethinking generalization

---

### Decision · Program_Chairs · 2023-01-20

**Decision:**

Reject

**Justification For Why Not Higher Score:**

NA

**Justification For Why Not Lower Score:**

NA

**Metareview: Summary, Strengths And Weaknesses:**

This work develops functional risk minimization problems as a generalization of empirical risk minimization. Examples for regression, classification, and value function estimation are given. However, links to noise augmentation of numerical search procedures (Langevin dynamics, simulated annealing) are not given, and the manner in which it can outperform existing results in terms of VC dimension or other quantitative learning-theoretic quantities are not given.

The main upshot seems to be in improved scalability of neural network training. However, the precise manner in which this improved scalability is actually achieved remains obscure. Due to these reasons, this work is below the bar of acceptance at this time, in accordance with the reviewer commentary.